# Transcriptomic Insights into the Response of the Olfactory Bulb to Selenium Treatment in a Mouse Model of Alzheimer’s Disease

**DOI:** 10.3390/ijms20122998

**Published:** 2019-06-19

**Authors:** Rui Zheng, Zhong-Hao Zhang, Yu-Xi Zhao, Chen Chen, Shi-Zheng Jia, Xian-Chun Cao, Li-Ming Shen, Jia-Zuan Ni, Guo-Li Song

**Affiliations:** Shenzhen Key Laboratory of Marine Bioresources and Ecology, College of Life Sciences and Oceanography, Shenzhen University, Shenzhen 518060, China; zhengrui_210@163.com (R.Z.); zhangzhonghao@email.szu.edu.cn (Z.-H.Z.); 13631337885@163.com (Y.-X.Z.); Chenchen430903@gmail.com (C.C.); jsz13602534834@163.com (S.-Z.J.); caoxianchun0306@outlook.com (X.-C.C.); slm@szu.edu.cn (L.-M.S.); jzni@szu.edu.cn (J.-Z.N.)

**Keywords:** selenomethionine, olfactory bulb, gene transcriptomics, differential gene expression

## Abstract

Alzheimer’s disease (AD) is a devastating neurodegenerative disorder characterized by the presence of extracellular senile plaques primarily composed of Aβ peptides and intracellular neurofibrillary tangles (NFTs) composed of hyperphosphorylated tau proteins. Olfactory dysfunction is an early clinical phenotype in AD and was reported to be attributable to the presence of NFTs, senile Aβ plaques in the olfactory bulb (OB). Our previous research found that selenomethionine (Se-Met), a major form of selenium (Se) in organisms, effectively increased oxidation resistance as well as reduced the generation and deposition of Aβ and tau hyperphosphorylation in the olfactory bulb of a triple transgenic mouse model of AD (3×Tg-AD), thereby suggesting a potential therapeutic option for AD. In this study, we further investigated changes in the transcriptome data of olfactory bulb tissues of 7-month-old triple transgenic AD (3×Tg-AD) mice treated with Se-Met (6 µg/mL) for three months. Comparison of the gene expression profile between Se-Met-treated and control mice revealed 143 differentially expressed genes (DEGs). Among these genes, 21 DEGs were upregulated and 122 downregulated. The DEGs were then annotated against the Gene Ontology (GO) and Kyoto Encyclopedia of Genes and Genomes (KEGG) databases. The results show that upregulated genes can be roughly classified into three types. Some of them mainly regulate the regeneration of nerves, such as *Fabp7*, *Evt5* and *Gal*; some are involved in improving cognition and memory, such as *Areg*; and some are involved in anti-oxidative stress and anti-apoptosis, such as *Adcyap1* and *Scg2*. The downregulated genes are mainly associated with inflammation and apoptosis, such as *Lrg1*, *Scgb3a1* and *Pglyrp1*. The reliability of the transcriptomic data was validated by quantitative real time polymerase chain reaction (qRT-PCR) for the selected genes. These results were in line with our previous study, which indicated therapeutic effects of Se-Met on AD mice, providing a theoretical basis for further study of the treatment of AD by Se-Met.

## 1. Introduction

Alzheimer’s disease (AD) is a neurodegenerative disease affecting a large population worldwide and is mainly characterized by progressive memory loss along with various other symptoms [1]. It involves nearly all the systems of the human body, such as the nervous system, endocrine system, and the immune system [2,3]. 

Studies have demonstrated that olfactory function decreases with aging, and the decline in olfaction is a common clinical phenotype of AD [4]. Olfactory dysfunction is prevalent in AD patients, and the functions of odor detection, discrimination, and identification are affected earlier than the decrease in cognitive ability [5]. Thus, olfactory dysfunction is considered a prodromal symptom of AD and has been proposed as a predicting factor of cognitive impairments [6]. The olfactory bulb is the first place that processes olfactory information received from the axons of olfactory receptor neurons [7]. It has been suggested that the olfactory bulb could be a starting point of pathology in AD [8]. In our previous study, we showed that selenomethionine (Se-Met), a major form of selenium in organisms, effectively increased oxidation resistance as well as reduced the production and deposition of Aβ and tau hyperphosphorylation in the hippocampus, cortex, and also the olfactory bulb of a triple transgenic mouse model of AD (3×Tg-AD) [9,10]. In addition, we determined the selenium levels in the tissues and organs of AD mice before and after treatment with Se-Met, and found that the level of selenium in brain tissue and olfactory bulb tissue increased significantly, indicating that selenium plays a biological role in the process of Se-Met treatment of AD mice [11]. Various mechanisms have been proposed to account for the therapeutic effects of Se-Met on AD, such as Se-Met-activated autophagy to eliminate Aβ and tau, promotion of neurogenesis in the hippocampus and reduction of inflammation and oxidative stress in the brain [9,12,13], but the foremost mechanism of the effect of Se-Met on the olfactory bulb is still unclear. 

Gene expression analysis is a useful tool in the study of neurodegenerative disease. To date, the majority of studies in the literature highlighted interesting genes on the basis of differential mRNA expression profile between control and experimental samples displaying an advantageous trait or the response to drug stimuli [14]. 

In this study, we investigated changes in the transcriptome data of olfactory bulb tissues in 7-month-old 3×Tg-AD mice after treatment by Se-Met for three months. We first searched for differentially expressed genes (DEGs) that were significantly altered (*p* < 0.05), followed by Gene Ontology (GO) analysis to find the specific function of the genes and Kyoto encyclopedia of genes and genomes (KEGG) analysis to bioassay enrichment of gene sets. To our knowledge, this is the first effort to study transcriptome sequences in the mouse olfactory bulb that may help understand the role of Se-Met in the treatment of AD. 

## 2. Results

### 2.1. Differential Gene Expression between 3×Tg-AD Mice and Se-Met-Treated 3×Tg-AD Mice

We selected the differently expressed genes according to the two criteria of (| log (fold change) |> 1) and the significance level (*p* < 0.05). The statistics of the number of significant differentially expressed genes are shown in Figure 1. One hundred forty-three genes were differentially expressed in the Se-Met-treated AD mice compared to the control mice, with 21 genes upregulated and 122 genes downregulated (Appendix A).

### 2.2. Overrepresentation Analysis

To determine the function of DEGs, a Gene Ontology (GO) enrichment analysis was performed (Table 1). GO results showed that DEGs were involved primarily in cellular responses to xenobiotic stimuli, flavonoid glucuronidation and glucuronate metabolic processes according to biological processes (BP). In terms of the cellular component (CC) annotation, “extracellular region”, “membrane-bounded vesicle” and “basolateral plasma membrane” showed large enrichment. Regarding the molecular function (MF) annotation, “transferase activity and transferring glycosyl groups”, “epidermal growth factor receptor binding”, “peptide hormone receptor binding” and “immunoglobulin binding” showed large enrichment.

To further determine which pathways could be directly affected by Se-Met treatment in 3xTg AD mice, DEGs were analyzed using the KEGG PATHWAY database. A Benjamini–Hochberg corrected *p*-value <0.05 was deemed significant. The DEGs were observed to be mainly part of signaling pathways including “metabolic”, “pentose and glucuronate interconversions”, “nicotinate and nicotinamide metabolism” and “ascorbate and aldarate metabolism” (Figure 2). A substantial number of other DEGs were also found to participate in different metabolic pathways such as “TNF signaling pathway” and “amino acids metabolism”.

To better clarify the genetic differences between Se-Met-treated and control AD mice, we selected the Top 20 upregulated and downregulated genes in the differential genes according to the *p*-value. The 20 genes upregulated with the highest fold-change in the Se-Met–treated group compared to the control group are shown in Table 2, and those downregulated are shown in Table 3.

To further investigate the effect of Se-Met on 3×Tg-AD mice, GO and KEGG analyses were performed on the top 20 up-regulated and the top 20 down-regulated genes, respectively. It was found that in the up-regulated genes, the biological processes most significantly affected by Se-Met are “positive regulation of secretion, neuropeptide signaling pathway and glial cell proliferation. In the cell components, Se-Met predominantly exerted its effects in the “extracellular region” and “neuron part”, etc. In molecular functions, Se-Met mainly affected “receptor binding” “G-protein coupled receptor binding” and “neuropeptide receptor binding” (Figure 3). In addition, in KEGG analysis, the “metabolism of cofactor and vitamins” signaling pathway was also significantly affected.

In the top 20 down-regulated genes, the biological processes “cell activation” and “response to cytokine” were most significantly affected. In the cell components, Se-Met mainly acted in the “extracellular region” and “extracellular space”. In the molecular functions, Se-Met mainly affected “receptor binding” “cell adhesion molecule binding” and “calcium-dependent protein binding” (Figure 4). In addition, through KEGG analysis, Se-Met mainly affected “transporters signal pathways” such as “cytokine–cytokine receptor interaction” and “nicotine addiction”.

### 2.3. Real-Time PCR Validation of Differentially Expressed Genes

To further verify the reliability of sequencing, nine genes were randomly selected from the top 20 upregulated and downregulated genes for qPCR experiments. The selections included upregulated genes, such as adenylate cyclase activating polypeptide 1 (*Adcyap1*), amphiregulin (*Areg*), secretogranin II (*Scg2*), fatty acid binding protein 7 (*Fabp7*), ets variant 5 (*Evt5*), and galanin (*Gal*), and downregulated genes, such as leucine-rich alpha-2-glycoprotein 1 (*Lrg*), secretoglobin family 3A member 1 (*Scgb3a1*), and peptidoglycan recognition protein 1 (*Pglyrp1*). The results further validated the result of deep sequencing (Figure 5).

## 3. Discussion

In the 7-month-old 3×Tg-AD mice, some of the early pathological features of AD began to appear, such as slightly decreased cognitive function, and the formation of Aβ [15]. Moreover, olfactory deficits also occurred before the clinical onset of cognitive deficits and coincided with AD pathology [16]. In our previous study [10], we found that, in addition to an improvement in cognitive ability, Se-Met treatment also induced a significant neuropathological improvement in the OB of 7-month-old 3×Tg-AD mice. Thus, AD-associated pathology in OB is also sensitive to pharmacological interventions, such as for those in the cortex and hippocampus [13]. To further clarify the effect of Se-Met treatment on neuropathology and behavioral changes at this time point, we carried out an OB transcriptome analysis, although using this method involves risks of identifying subtle changes at the level of local gene expression that may not be representative of whole brain gene expression.

We found in the GO and KEGG analysis of the total differential genes, “transferase activity and transferring glycosyl groups”, “epidermal growth factor receptor binding”, “peptide hormone receptor binding” and “immunoglobulin binding” showing large enrichment. The DEGs were observed to be mainly involved in signaling pathways including “metabolic”, “pentose and glucuronate interconversions” and “ascorbate and aldarate metabolism”. Many patients with Alzheimer’s disease have associated vascular disease [17], including changes in blood rheology: hyperglycemia, abnormal platelet function, and elevated blood coagulation factors, resulting in increased blood viscosity [18,19]. At the same time, pathological changes, including capillary basement membrane thickening, endothelial cell swelling and hyperplasia [20], thickening of the wall, glycoprotein deposition [20,21], which causes ischemia and hypoxia of nerve tissue, also lead to nerve tissue necrosis and dysfunction. We performed GO analysis of the total differential genes and found that Se-Met has a certain effect on “epidermal growth factor receptor binding” and “peptide hormone receptor binding”, which suggests a potential beneficial effect on vascular endothelial cells. Vascular endothelial cells are the main cellular components that constitute the vascular wall and the blood-brain barrier [22]. They secrete various vasoactive substances, including endothelin, nitric oxide, and prostaglandins, which play important roles in the processes of eliminating brain wastes [23]. Thus, the improvement of vascular endothelial cells might also play a role in the therapeutic effect of Se-Met on AD.

The brain is the organ with the highest cholesterol content and contains 25% of the unesterified cholesterol in the body [24]. Due to the presence of the blood-brain barrier, almost all cholesterol in the brain is self-synthesized [25]. Neurons are not able to efficiently synthesize cholesterol, and the synthesis of cholesterol is mainly dependent on astrocytes [26]. Most of the cholesterol is transported to neurons by APOE (apolipoprotein E) containing lipoproteins [27]. The increase of cholesterol level in the neuronal cell leads to the increase of β-secretase activity and Aβ generation [28,29], the phosphorylation of Tau [30,31], and a reduced level of synaptophysin [32]. Through the enrichment of KEGG, we found that Se-Met could effectively improve the lipid metabolism in 3×Tg-AD mice. Therefore, we speculated that the improving effect of Se-Met on lipid metabolism might contribute to its effect of protecting neurons and ameliorating the cognitive deficit in AD mice.

### 3.1. Upregulated Genes in 3×Tg-AD (Se-Met) Mice

This work has identified several key genes upregulated after treatment with Se-Met, including Adcyap1, which encodes a member of the glucagon superfamily of hormones that have important roles in the pathogenesis of AD [33]. Pituitary adenylate cyclase activating polypeptide (*PACAP*) is a protein encoded by the Adcypa1 gene that is expressed primarily in sensory neurons, sympathetic preganglionic neurons, and parasympathetic ganglionic neurons. *PACAP* has been shown to function as a neuro-hormone, a neurotransmitter, and a neurotrophic factor [34]. It has been demonstrated that *PACAP* levels start to decline before the onset of AD dementia [35]. This reduction in *PACAP* level is region specific, targeting vulnerable areas in the AD brain [36]. In addition, *PACAP* binds to PAC1 receptors and activates adenylate cyclase (AC)-linked signal transduction pathways in cells. *PACAP* triggers anti-apoptotic transcriptional target gene expression and inhibits apoptotic signaling responses, including ROS generation, mitochondrial Bax and cytochrome C release, and subsequent caspase-3 activation [37,38].

Another upregulated DEG, *Areg,* encodes a member of the epidermal growth factor (EGF) family that is proteolytically processed to generate a mature protein [39]. EGF was reported to prevent Aβ-induced damage to the cerebrovascular system, thus implicating angiogenic pathways as potential therapeutic targets for AD [40]. The *Areg*-encoded protein is a ligand of the epidermal growth factor receptor (EGFR) and has been shown to play a role in immunity, inflammation, tissue repair, and lung and mammary gland development [41]. It was reported that EGFR expression was significantly increased in the OB of AD and that EGFR signaling was necessary for olfactory learning and discrimination [42].

*Scg2* is a secreted neuroendocrine marker observed in prostatic small-cell neuroendocrine carcinoma [43]. It has been demonstrated that the levels of the *Scg2*-derived peptide secretoneurin decreased in cerebrospinal fluid (CSF) in AD mice [43]. Secretoneurin induces dopamine release from nigrostriatal neurons, influences neurite outgrowth, exerts chemotactic effects on monocytes, eosinophils and endothelial cells, and has angiogenic effects [44]. Moreover, *Scg2* was also reported to be involved in many biological processes, such as induction of MAPK cascade and negative regulation of extrinsic apoptotic signaling pathway [45,46]. Therefore, Se-Met might exert its anti-AD effects through inhibition of apoptotic signaling responses, activation of EGFR signaling and increasing the secretoneurin level.

*Evt5* plays an important role in cell proliferation and differentiation [47], such as positive regulation of glial cell proliferation, regulation of DNA transcription, and cellular response to oxidative stress [48,49]. *Fabp7* is known as a brain lipid-binding protein. It has been demonstrated that the synthesis of *Fabp7* by astrocytes in neurogenic niches has implications for neurogenesis in an AD model [50]. Moreover, both upstream modulators of *Fabp7* expression including transcription factors Notch and Pax6 and downstream mediators of *Fabp7* such as fatty acid-binding proteins, were reported to enhance the cellular uptake of retinoic acid, modulate gene transcription and promote nerve regeneration [51,52]. In AD, the disruption of neurogenesis results in the loss of normal neural function and the inability to generate a sufficient number of neurons, which ultimately leads to brain dysfunction [53]. Induction of neurogenesis can stimulate self-repair mechanisms in the brain to repair brain regions suffering from degenerative damage. There are a large number of regenerative and differentiated neural stem cells in OB [54]. Our previous research found that Se-Met promotes neurogenesis in the hippocampus [12]. Therefore, we hypothesize that Se-Met could enhance *Evt5* and *Fabp7* expression and promote neuroregeneration in OB.

Galanin (*Gal*) is an endogenously expressed neuropeptide [55] and is known to be involved in several prominent processes of the nervous system, such as memory function, regulation of hypothalamopituitary hormones, inflammation and neuroprotection [56,57]. *Gal* has been reported to have neuroprotective roles in several cases, such as traumatic brain injury, β-amyloid poisoning, glutamate-induced excitotoxicity and high-glucose-induced apoptosis [58]. The upregulation of *Gal* and endocannabinoid systems supported the hypothesis of their neuroprotective roles, which are established prior to the onset of clear clinical cognitive symptoms of the disease [59].

### 3.2. Downregulated Genes in 3×Tg-AD (Se-Met) Mice

*Lrg1* is reported to be highly elevated in idiopathic normal pressure hydrocephalus (iNPH) and is believed to be an iNPH-specific candidate biomarker in cerebro-spinal fluid (CSF) [60,61]. Cognitive impairment due to iNPH initially presents as frontal lobe dysfunction with deficits in attention, execution, and thinking. In an advanced stage of iNPH, short-term memory is impaired like that in AD. Differentiating iNPH from Alzheimer’s disease (AD) is difficult when the cognitive function of the patient is highly impaired [61]. Therefore, we speculate that Lrg1 has a similar function in AD. It is also reported that the expression level of *Lrg1* is correlated with inflammation [62]. Therefore, a significant decrease in the *Lrg1* expression level after Se-Met treatment of 3xTg AD mice confirmed that Se-Met has a role in alleviating the inflammation of AD, which agrees with our previous results [9,10].

*Pglyrp1* (peptidoglycan recognition protein 1) is a neutrophil granule protein with antibacterial properties, and when multimerized, it becomes a functional ligand of a triggering receptor expressed on myeloid cells (TREM-1), a cell-surface receptor involved in innate immune activation and amplification of inflammatory response to bacterial infections [63]. In addition, *Pglyrp1* was also reported to be dysregulated in AD patient sera samples and could be a potential serum biomarker for the identification of AD [64].

*Scgb3a1* encodes a small secreted protein that plays a role in inhibiting cell growth [65]. Spliced transcript variants encoding multiple isoforms have been observed for this gene [66]. *Sc*gb3a1 is a suggested tumor suppressor gene that inhibits cell growth and invasion and is methylated and down-regulated in many epithelial cancers [67]. The study found that phosphorylation of Akt in cancer can effectively inhibit the expression of *Sc*gb3a1 [68]. Se-Met significantly activated Akt expression in the brain of AD mice [11]. It is speculated that Se-Met might reduce the expression of *Sc*gb3a1 by activating Akt in AD mice.

In summary, we provide comprehensive insights into the transcriptome of 3×Tg-AD mice’s OB tissues. The results showed that upregulated genes mainly regulate the regeneration of nerves, such as *Fabp7*, *Evt5* and *Gal*; improve cognition and memory, such as *Areg*; and impact oxidative stress and apoptosis, such as *Adcyap1* and *Scg2*. The downregulated genes are mainly associated with inflammation and apoptosis, such as *Lrg1*, *Scgb3a1* and *Pglyrp1*. This study is, to the best of our knowledge, the first report of Se-Met-treated OB whole transcriptome sequencing (RNA-Seq). Our results revealed a number of genes implicated in AD pathology and drug metabolism after the treatment with Se-Met. This is in line with previously described effects of Se-Met on AD model mice and provides a theoretical basis for further study of the treatment of AD by Se-Met.

## 4. Material and Methods 

### 4.1. Transgenic Mice 

3×Tg-AD mice (4 months old), purchased from the Jackson Laboratory (Bar Harbor, Maine, ME, USA), were kept with accessible food and water under a 12-h light/dark cycle [69]. All animal experiments and procedures were approved by the Ethics Committee of Shenzhen University (Permit Number: AEWC-20140615-002, 15 July 2014). Mice (*n* = 6; 6 males and 6 females) were treated with 6 μg/mL Se-Met (Sigma-Aldrich, Santa Clara, CA, USA) in drinking water or received normal drinking water for 3 months.

### 4.2. Tissue Processing

Mice were euthanized after treatment with Se-Met for 90 days, and the brains were rapidly removed. One hemibrain was washed with PBS in 4 °C, and the olfactory bulb was isolated. Total RNA was extracted from the dissected olfactory bulb by the RNeasy Micro Kit (74004, Qiagen, Beijing, China) according to the manufacturer’s instructions and eluted with 14 mL of RNase-free water. RNA concentrations were determined using NanoDrop 1000 (Thermo Scientific, Shanghai, China). The other hemibrain was snap frozen and stored at −80 °C until further use.

### 4.3. RNA-Seq Analysis

The following parameters were set for RNA quality control: A260/A280 ratio >1.8, A260/A230 ratio >2.0 and RIN value >7.0. The TruSeq RNA sample preparation kit (Illumina, San Diego, CA, USA) was used for RNA-seq library preparation. Finally, sequencing of the libraries was conducted on an Illumina HiSeq^TM^ 2500 system. A computational pipeline was used to process RNA-seq data. Sequence data were mapped to the mouse reference genome GRCm38 with Tophat v1.4.0 [70] with default parameters. HTSeq-count was subsequently employed to convert aligned short reads into read counts for each gene model in the European Bioinformatics Institute (ENSEMBL) database release [71]. Differential expression was assessed by DEseq using read counts as input [72]. The Benjamini–Hochberg multiple test correction method was enabled. Differentially expressed genes were chosen according to the criteria fold change >2 and adjusted *p*-value <0.05.

### 4.4. Differential Expression Analysis

To normalize for GC-content within and between lanes, full-quantile normalization as implemented in the EDASeq R package was performed. Dispersion was estimated with the help of the DESeq R package, treating all samples as replicates of a single condition [73]. Subsequent analyses were based on fitted dispersion values. Two conditions were compared with a negative binomial test. Differential expression was deemed significant if the Benjamini–Hochberg corrected *p*-value was less than 0.05. Se-Met-treated versus control gene expressions were compared. Overlap in the up- and downregulated genes between the different groups was visualized using BioVenn [74,75].

Functional annotation clustering was performed using the DAVID software tool [76] or OmicsBean (http://www.omicsbean.cn) database. Annotations from Gene Ontology (GO) [77] and Kyoto Encyclopedia of Genes and Genomes (KEGG) [78] were tested for enrichment.

### 4.5. Real-Time PCR

Ten genes were selected for further real-time PCR analysis, including *Adcyap1* (adenylate cyclase activating polypeptide 1), *Areg* (amphiregulin), *Scg2* (secretogranin II), *Fabp7* (fatty acid binding protein 7), *Etv5* (ets variant 5), *Gal* (galanin), GAPDH (glyceraldehyde-3-phosphate dehydrogenase), *Lrg* (leucine-rich alpha-2-glycoprotein 1), *Scgb3a1* (secretoglobin, family 3A, member 1), and *Pglyrp1*(peptidoglycan recognition protein 1).

Complementary DNA was produced from cellular RNA (5 μg) using a Prime Script TM RT reagent Kit (Perfect Real Time) (Takara, Shanghai, China). Real-time PCR primers were designed using PRIMER EXPRESS software (Version 1.5, Applied Biosystems), and the specificity of sequences was verified using BLAST (http://www.ncbi.nlm.nih.gov/BLAST/). Reactions were performed in 10 μL quantities of diluted cDNA sample, primers (100, 200, or 300 nM), and a SYBR Green PCR Master Mix containing nucleotides, AmpliTaq Gold DNA polymerase, and optimized buffer components (Applied Biosystems). Primers were purchased from BGI (Beijing Genomics Institute, Beijing, China) as intron-spanning validated primer pairs. Reactions were assayed using an Applied Biosystems Prism 7700 sequence detection system. After cycling, a melting curve was produced via the slow denaturation of PCR end products to validate amplification specificity. Predicted cycle threshold (CT) values were exported into EXCEL worksheets for analysis. Comparative CT methods were used to determine relative gene expression folds to GAPDH. GAPDH primers were used to confirm gene expression levels. The primers used for real-time PCR were as Table 4:

### 4.6. Statistical Analysis

The data were analyzed using GraphPad Prism software. All data were expressed as the mean ± SEM and considered statistically significant at a level of *p* < 0.05. A two-way *t*-test was used to analyze the QPCR analysis data.

## Figures and Tables

**Figure 1 ijms-20-02998-f001:**
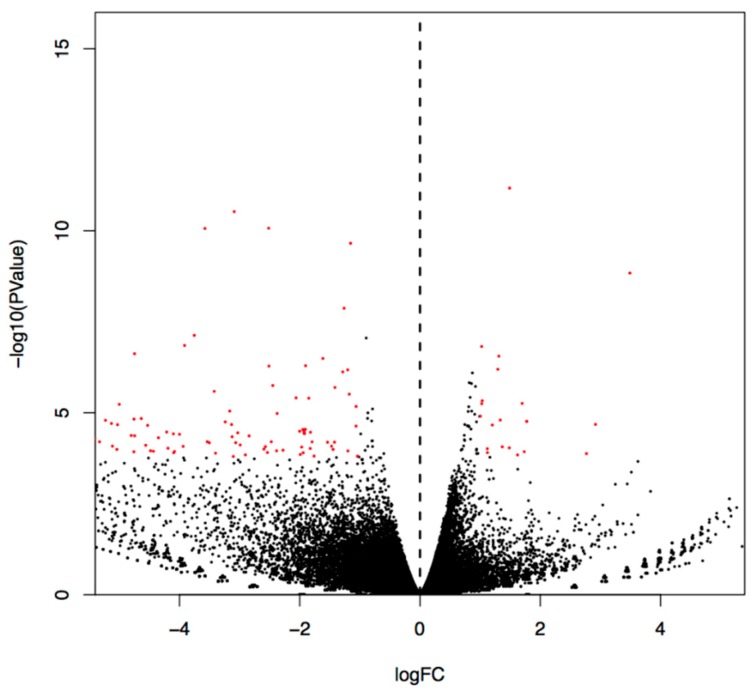
Volcano plots of significantly differentially expressed genes. log FC (fold-change) (X-axis) and p-value (Y-axis) of differentially expressed genes (DEGs) comparing control versus Se-Met-treated 3×Tg-AD mice. Each dot represents a single DEG, and a red dot illustrates statistical significance (*p* = 0.05). Left-downregulated; Right–upregulated.

**Figure 2 ijms-20-02998-f002:**
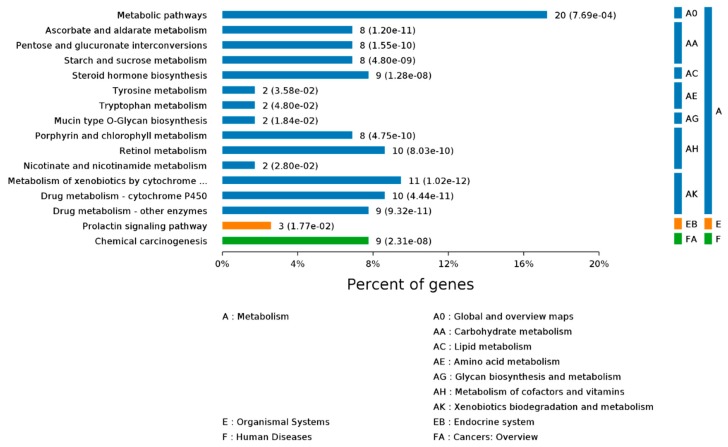
KEGG Pathway names for DEGs. Based on the KEGG biology pathway database (http://www.genome.jp/) from the perspective of complex regulatory networks, the differential gene sets were enriched, thus the most relevant biological pathways on the differentially expressed gene were extracted.

**Figure 3 ijms-20-02998-f003:**
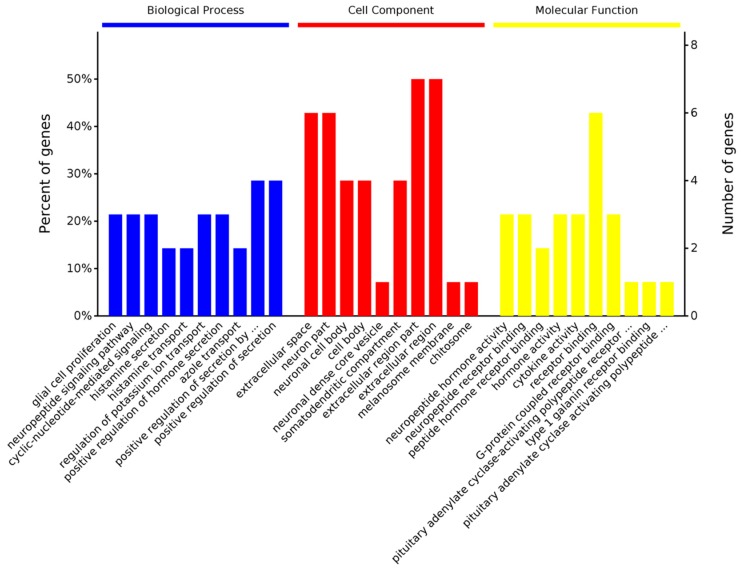
Gene Ontology identifiers in the cluster of overlapping differentially expressed top 20 genes upregulated in 3xTg AD mice (Se-Met-treated) compared to 3×Tg AD mice.

**Figure 4 ijms-20-02998-f004:**
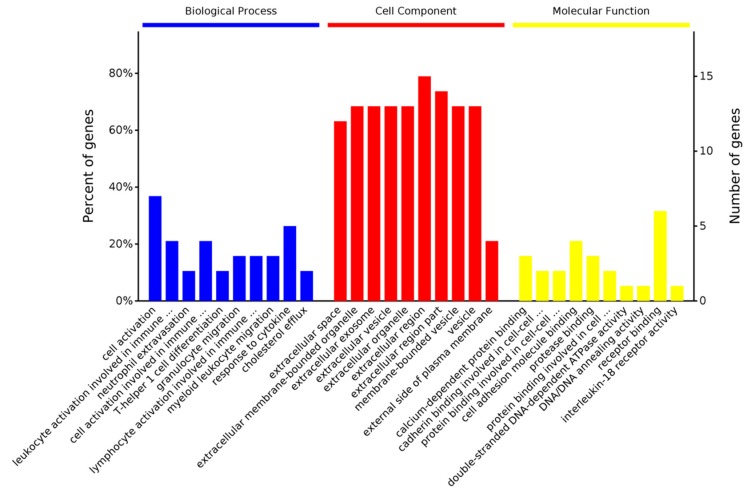
Gene Ontology identifiers in the cluster of overlapping differentially expressed top 20 genes downregulated in 3×Tg-AD mice (Se-Met-treated) compared to 3×Tg-AD mice.

**Figure 5 ijms-20-02998-f005:**
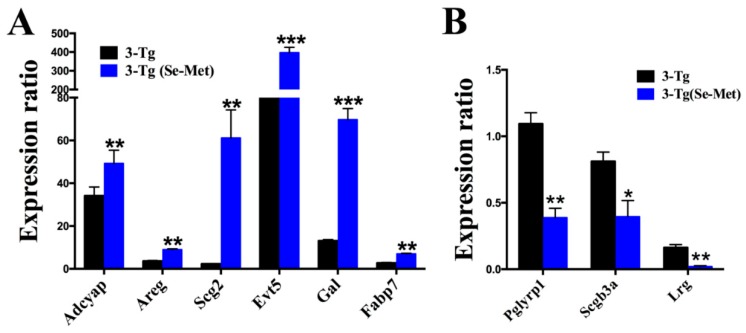
Validation of sequencing results for upregulated (**A**) or downregulated genes (**B**) using qRT-PCR. The housekeeping gene GAPDH was used for normalization. Significantly differential expression is indicated as *** *p* < 0.001; ** *p* < 0.01; * *p* < 0.05.

**Table 1 ijms-20-02998-t001:** A list of the Gene Ontology identifiers in the cluster of overlapping differentially expressed genes in Se-Met-treated AD mice compared to control AD mice.

Category	GO Class	Function	Sample Number	Background Number	*p* Value	Gene Names
BP	GO:0052697	xenobiotic glucuronidation	4	5	2.44 × 10^−7^	Ugt1a2; Ugt1a1; Ugt1a10; Ugt1a6a
BP	GO:0052696	flavonoid glucuronidation	4	9	1.36 × 10^−6^	Ugt1a2; Ugt1a1; Ugt1a10; Ugt1a6a
BP	GO:0009813	flavonoid biosynthetic process	4	9	1.36 × 10^−6^	Ugt1a2; Ugt1a1; Ugt1a10; Ugt1a6a
BP	GO:0052695	cellular glucuronidation	4	10	1.89 × 10^−6^	Ugt1a2; Ugt1a1; Ugt1a10; Ugt1a6a
BP	GO:0019585	glucuronate metabolic process	4	11	2.56 × 10^−6^	Ugt1a2; Ugt1a1; Ugt1a10; Ugt1a6a
BP	GO:0006063	uronic acid metabolic process	4	11	2.56 × 10^−6^	Ugt1a2; Ugt1a1; Ugt1a10; Ugt1a6a
BP	GO:0009812	flavonoid metabolic process	4	11	2.56 × 10^−6^	Ugt1a2; Ugt1a1; Ugt1a10; Ugt1a6a
BP	GO:0006805	xenobiotic metabolic process	5	54	4.93 × 10^−5^	Ugt1a2; Ugt1a1; Cyp2f2; Ugt1a10; Ugt1a6a
BP	GO:0071466	cellular response to xenobiotic stimulus	5	58	6.77 × 10^−5^	Ugt1a2; Ugt1a1; Cyp2f2; Ugt1a10; Ugt1a6a
BP	GO:0009410	response to xenobiotic stimulus	5	64	1.04 × 10^−4^	Ugt1a2; Ugt1a1; Cyp2f2; Ugt1a10; Ugt1a6a
CC	GO:0005615	extracellular space	25	1011	7.36 × 10^−8^	Pigr; Adcyap1; Tacstd2; Fetub; Pglyrp1; Gpx3; Sftpd; Dmbt1; Pon1; Ces1d; Lbp; Scgb3a1; Mup4; C1s; Bpifb1; Areg; Msln; Anxa1; Tnfsf15; Serpinb11; Bpifa1; Nppa; Il1r1; Scg2; St14
CC	GO:0005576	extracellular region	52	3593	6.55 × 10^−5^	Pigr; Reg3g; Bpifa1; Tspan1; Tacstd2; Krt14; Ccdc3; Vwf; Fetub; Pglyrp1; Gpx3; Scnn1a; Krt5; Sftpd; Dmbt1; Ces1d; Pon1; Agr2; Obp2a; Cd44; Sult1c1; Lbp; Chi3l4; Scgb3a1; Mup4; Wfdc18; C1s; Gsta3; Muc20; Il1r1; Bpifb1; Krt18; Areg; Wfdc2; Msln; Lypd2; Anxa1; Tnfsf15; Epcam; Serpinb11; Steap4; Slc5a9; Cd177; Clic6; Adcyap1; Ugt1a6a; St14; Tmprss2; Slc44a4; Slc39a4; Scg2; Nppa
CC	GO:0044421	extracellular region part	44	3071	5.7 × 10^−6^	Pigr; Adcyap1; Tspan1; Tacstd2; Krt14; Vwf; Fetub; Pglyrp1; Gpx3; Scnn1a; Krt5; Sftpd; Dmbt1; Ces1d; Pon1; Cd44; Sult1c1; Lbp; Scgb3a1; Mup4; C1s; Gsta3; Il1r1; Bpifb1; Krt18; Areg; Wfdc2; Msln; Anxa1; Tnfsf15; Epcam; Serpinb11; Steap4; Slc5a9; Cd177; Clic6; Bpifa1; Ugt1a6a; St14; Tmprss2; Slc44a4; Slc39a4; Scg2; Nppa
CC	GO:0016323	basolateral plasma membrane	8	176	3.91 × 10^−5^	Tacstd2; Cd44; St14; Cldn8; Epcam; Cldn7; Muc20; Aqp3
CC	GO:0070062	extracellular vesicular exosome	32	2280	1.4 × 10^−4^	Pigr; Pglyrp1; Tspan1; Tacstd2; Krt14; Gsta3; Fetub; Scnn1a; Krt5; Dmbt1; Pon1; Cd44; Sult1c1; Lbp; Scgb3a1; Gpx3; C1s; Vwf; Bpifb1; Krt18; Wfdc2; Anxa1; Epcam; Steap4; Slc5a9; Cd177; Clic6; Ugt1a6a; Tmprss2; Slc44a4; Slc39a4; St14
CC	GO:1903561	extracellular vesicle	32	2280	1.41 × 10^−4^	Pigr; Pglyrp1; Tspan1; Tacstd2; Krt14; Gsta3; Fetub; Scnn1a; Krt5; Dmbt1; Pon1; Cd44; Sult1c1; Lbp; Scgb3a1; Gpx3; C1s; Vwf; Bpifb1; Krt18; Wfdc2; Anxa1; Epcam; Steap4; Slc5a9; Cd177; Clic6; Ugt1a6a; Tmprss2; Slc44a4; Slc39a4; St14
CC	GO:0043230	extracellular organelle	32	2284	1.44 × 10^−4^	Pigr; Pglyrp1; Tspan1; Tacstd2; Krt14; Gsta3; Fetub; Scnn1a; Krt5; Dmbt1; Pon1; Cd44; Sult1c1; Lbp; Scgb3a1; Gpx3; C1s; Vwf; Bpifb1; Krt18; Wfdc2; Anxa1; Epcam; Steap4; Slc5a9; Cd177; Clic6; Ugt1a6a; Tmprss2; Slc44a4; Slc39a4; St14
CC	GO:0065010	extracellular membrane-bounded organelle	32	2284	1.45 × 10^−4^	Pigr; Pglyrp1; Tspan1; Tacstd2; Krt14; Gsta3; Fetub; Scnn1a; Krt5; Dmbt1; Pon1; Cd44; Sult1c1; Lbp; Scgb3a1; Gpx3; C1s; Vwf; Bpifb1; Krt18; Wfdc2; Anxa1; Epcam; Steap4; Slc5a9; Cd177; Clic6; Ugt1a6a; Tmprss2; Slc44a4; Slc39a4; St14
CC	GO:0031988	membrane-bounded vesicle	35	2756	4.34 × 10^−4^	Pigr; Pglyrp1; Tspan1; Tacstd2; Krt14; Th; Gsta3; Fetub; Gpx3; Scnn1a; Krt5; Dmbt1; Pon1; Cd44; Sult1c1; Lbp; Scgb3a1; Scg2; C1s; Vwf; Bpifb1; Krt18; Wfdc2; Galnt15; Anxa1; Epcam; Steap4; Slc5a9; Cd177; Clic6; Ugt1a6a; Tmprss2; Slc44a4; Slc39a4; St14
CC	GO:0098590	plasma membrane region	10	428	8.69 × 10^−4^	Aqp3; Cd44; St14; Cldn8; Slc39a4; Scnn1a; Epcam; Cldn7; Muc20; Tacstd2
MF	GO:0015020	glucuronosyltransferase activity	5	74	1.99 × 10^−4^	Ugt1a2; Ugt1a1; Galnt15; Ugt1a6a; Ugt1a10
MF	GO:0008194	UDP-glycosyltransferase activity	6	135	4.03 × 10^−4^	Ugt1a2; Galnt15; Ugt1a1; Ugt1a6a; Gal; Ugt1a10
MF	GO:0016758	transferase activity, transferring hexosyl groups	6	165	1.11 × 10^−4^	Ugt1a2; Galnt15; Ugt1a1; Ugt1a6a; Gal; Ugt1a10
MF	GO:0005154	epidermal growth factor receptor binding	3	30	1.4 × 10^−3^	Pigr; Cd44; Areg
MF	GO:0016757	transferase activity, transferring glycosyl groups	7	243	1.62 × 10^−3^	St6galnac1; Ugt1a2; Galnt15; Ugt1a1; Ugt1a6a; Gal; Ugt1a10
MF	GO:0001972	retinoic acid binding	2	12	3.85 × 10^−3^	Ugt1a2; Ugt1a1
MF	GO:0019865	immunoglobulin binding	2	14	5.03 × 10^−3^	Vwf; Fcamr
MF	GO:0038024	cargo receptor activity	3	53	6.34 × 10^−3^	Ildr1; Tmprss2; Dmbt1
MF	GO:0051428	peptide hormone receptor binding	2	16	6.36 × 10^−3^	Adcyap1; Nppa
MF	GO:0033293	monocarboxylic acid binding	3	55	6.9 × 10^−3^	Ugt1a2; Ugt1a1; Fabp7

BP, biological process; CC, cellular component; MF, molecular function.

**Table 2 ijms-20-02998-t002:** 20 genes upregulated in Se-Met-treated 3×Tg-AD mice versus 3×Tg-AD mice.

Gene Name	Gene Description	logFC	*p*_Value
*Adcyap1*	adenylate cyclase activating polypeptide 1	1.49	6.74 × 10^−12^
*Areg*	amphiregulin	3.49	1.46 × 10^−9^
*Scg2*	secretogranin II	1.02	1.51 × 10^−7^
*Th*	tyrosine hydroxylase	1.31	2.8 × 10^−7^
*Fabp7*	fatty acid binding protein 7	1.29	6.41 × 10^−7^
*Pip5k1b*	phosphatidylinositol-4-phosphate 5-kinase, type 1 beta	1.03	4.68 × 10^−6^
*Trhr2*	thyrotropin releasing hormone receptor 2	1.69	5.55 × 10^−6^
*Etv5*	ets variant 5	1.02	5.68 × 10^−5^
*fantom3_1110005E01*	///	1.01	1.25 × 10^−5^
*Col28a1*	collagen, type XXVIII, alpha 1	1.33	1.59 × 10^−5^
*fantom3_D930027K06*	///	1.77	1.73 × 10^−5^
*fantom3_D930001D16*	///	1.77	1.73 × 10^−5^
*Gm10635*	predicted gene 10635	2.91	2.09 × 10^−5^
*Nppa*	natriuretic peptide type A	1.20	2.18 × 10^−5^
*Nt5c1a*	5′-nucleotidase, cytosolic IA	1.36	8.67 × 10^−5^
*Gal*	galanin	1.48	9.23 × 10^−5^
*fantom3_A430024L20*	///	1.12	9.76 × 10^−5^
*Rxfp1*	relaxin/insulin-like family peptide receptor 1	1.73	1.18 × 10^−4^
*fantom3_C330006P03*	///	1.12	1.25 × 10^−4^
*Tnfsf15*	tumor necrosis factor (ligand) superfamily, member 15	2.76	1.33 × 10^−4^

Note: “///” indicates that there is currently no relevant research.

**Table 3 ijms-20-02998-t003:** Top 20 genes downregulated in Se-Met-treated 3×Tg-AD mice versus 3×Tg-AD mice.

Gene Name	Gene Description	logFC	*p*_Value
*Lrg1*	leucine-rich alpha-2-glycoprotein 1	−3.08	2.99 × 10^−11^
*Anxa1*	annexin A1	−2.51	8.52 × 10^−11^
*Cd177*	CD177 antigen	−3.57	8.66 × 10^−11^
*Vwf*	Von Willebrand factor	−1.15	2.21 × 10^−10^
*Il1r1*	interleukin 1 receptor, type I	−1.26	1.35 × 10^−9^
*Scgb3a1*	secretoglobin, family 3A, member 1	−3.75	7.5 × 10^−8^
*fantom3_1110030K16*	///	−3.91	1.42 × 10^−7^
*Abca13*	ATP-binding cassette, sub-family A (ABC1), member 13	−4.74	2.4 × 10^−7^
*Pglyrp1*	peptidoglycan recognition protein 1	−1.61	3.23 × 10^−7^
*Bpifb1*	BPI fold containing family B, member 1	−6.70	4.13 × 10^−7^
*Gpx3*	glutathione peroxidase 3	−1.90	5.13 × 10^−7^
*Dlk1*	delta-like 1 homolog (Drosophila)	−2.51	5.23 × 10^−7^
*Gabrp*	gamma-aminobutyric acid (GABA) A receptor, pi	−6.28	5.51 × 10^−7^
*Serpina3n*	serine (or cysteine) peptidase inhibitor, clade A, member 3N	−1.20	6.65 × 10^−7^
*S100a11*	S100 calcium binding protein A11	−1.28	7.58 × 10^−7^
*Muc5b*	mucin 5, subtype B, tracheobronchial	−6.35	7.95 × 10^−7^
*Lypd2*	Ly6/Plaur domain containing 2	−5.61	8.2 × 10^−7^
*Pon1*	paraoxonase 1	−7.07	9.8 × 10^−7^
*Il18r1*	interleukin 18 receptor 1	−2.44	1.79 × 10^−6^
*Ccdc3*	coiled-coil domain containing 3	−1.41	2.03 × 10^−6^

Note: “///” indicates that there is currently no relevant research.

**Table 4 ijms-20-02998-t004:** Primers sequences information.

Genes Name		Sequence Information
*Adcyap1*	Forward	5′-GGAGAAAAGTGGAGGGAGCA-3′
	Reverse	5′-TGTCTATACCTTTTCCCAAGGACTG-3′
*Areg*	Forward	5′-TGGATTGGACCTCAATGACA-3′
	Reverse	5′-AGCCAGGTATTTGTGGTTCG-3′
*Scg2*	Forward	5′-CCAATGGTCATGGTATTGACA-3′
	Reverse	5′-TTTGCTCCAGAACTCCACAA-3′
*Fabp7*	Forward	5′-AGGCTTTCTGTGCTAC-3′
	Reverse	5′-ATTACCGTTGGTTTGG-3′
*Evt5*	Forward	5′-GAAGTGCCTAACTGCCAGTCACCC-3′
	Reverse	5′-GGCACCACGCAAGTGTCATCGA-3′
*Gal*	Forward	5′-CACTCTGGGACTTGGGATG-3′
	Reverse	5′-CAGGC AAGAGGGAGTTACAA-3′
GAPDH	Forward	5′-CAGCCTCGTCTCATAGACAAGATG-3′
	Reverse	5′-CAATGTCCACTTTGTCACAAGAGAA-3′
*Lrg*	Forward	5′-GGAGCAGCTATGGTCTCTTG-3′
	Reverse	5′-AGTATCAGGCATTCCTTGAG-3′
*Scgb3a1*	Forward	5′-CTCAACCCGCTGAAGCTC-3′
	Reverse	5′-CTTCTGGGAGCCCTCTATGA-3′
*Pglyrp1*	Forward	5′-GTGGTGATCTCACACACAGC-3′
	Reverse	5′-GTGTGGTCACCCTTGATGTT-3′

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
