# Peer review of "Transcriptomic Insights into the Response of the Olfactory Bulb to Selenium Treatment in a Mouse Model of Alzheimer’s Disease"

_ijms, 2019, doi:10.3390/ijms20122998_

Round 1
Reviewer 1 Report
The authors used RNA-seq to study the effect of Se-Met treatment on OB gene expression in an AD mouse model. In particular, the authors identified ~100 differentially expressed genes, analyzed their gene ontology, and validated 9 genes via RT-qPCR. Both the experimental design and data analysis are sound; and the results provide mechanistic explanations to Se-Met's effect on AD mouse models. Therefore, this manuscript is suitable for publication after addressing a few minor comments:
In 2.3, "the mouse genome rn5" is probably a typo, because rn5 is a rat genome.
The authors used "log FC", which I assume is natural log, in most figures and tables, but "log2 FC" in the text of 3.1. These should be unified.
In Table 2, gene names should not be broken between different lines.
In discussion, the authors only mentioned "4-month-old 3xTg AD mice". They should clarify that the treatment starts at 4-month-old, but the assay was performed after the 3-month treatment, making the mice 7-month-old when RNA is sequenced.
Author Response
Response to Reviewer 1
Dear Reviewers,
Thank you for you thoughtful, helpful, and most kind review of manuscript ijms-519761. Your comments and suggestions have been incorporated into the revised draft. Specific revisions are note below.
Question 1: In 2.3, "the mouse genome rn5" is probably a typo, because rn5 is a rat genome.
Answer:
We thank the reviewer for your careful checking. The correct description should be GRCm38 and we revised the description in the text of 2.3. Thank you.
Question 2: The authors used "log FC", which I assume is natural log, in most figures and tables, but "log2 FC" in the text of 3.1. These should be unified.
Answer: We thank the reviewer for yourkind suggestion.We have checked and revised accordingly in the text of 3.1.
Question 3: In Table 2, gene names should not be broken between different lines.
Answer: We thank the reviewer for yourkind suggestion. We have revised the description in Table 2 accordingly. Thank you.
Question 4: In discussion, the authors only mentioned "4-month-old 3xTg AD mice". They should clarify that the treatment starts at 4-month-old, but the assay was performed after the 3-month treatment, making the mice 7-month-old when RNA is sequenced.
Answer: We have revised the description as “7-month-old 3xTg AD mice” in discussion. Thank you.
Thank you again for your kind and thoughtful comments. We hope that the revision addresses you concerns.
Yours sincerely,
Guo-Li Song
On behalf of all authors
College of Life Sciences and Oceanography
Shenzhen University
Shenzhen, 518071
P.R. China
Reviewer 2 Report
The authors have already demonstrated that the treatment of 3xtgmice with Selenomethionine is able to have neuroprotective effects. The present work investigates, through transcriptome analysis, the changes in gene expression in the olfactory bulb of these mice. The work is well-conducted and properly discussed. Data are correctly analysed and presented. References are complete. I recommend the publication of the article after a few small revisions.
In the discussion paragraph:
-line55: when citing "our previous study", just add the number of the reference.
-line 65-67: we found that...or use showing (instead of showed)
-line 69: try avoiding the use of "and so on".
-line 80: which (not Which)
-line 88: increased is not correct. Revise the phrase.
-line 141-143: this phrase is not clear to me.
I would recommend putting the general conclusions at the very end of the article and not before the two specific commentaries on DEGs.
Author Response
Response to Reviewer 2
Dear Reviewers,
Thank you for you thoughtful, helpful, and most kind review of manuscript ijms-519761. Your comments and suggestions have been incorporated into the revised draft. Specific revisions are note below.
Question 1: line55: when citing "our previous study", just add the number of the reference.
Answer:
We thank the reviewer for your kind suggestion. The number of the referencewas added inthe discussion (lines 55).
Question 1:line 65-67: we found that...or use showing (instead of showed)
Answer: We thank the reviewer for yourkind suggestion. We have revised the description in the discussion (lines 65-67).
Question 2: line 69: try avoiding the use of "and so on".
Answer: We have deleted the description in the discussion (lines 69) accordingly. Thank you.
Question 3: -line 80: which (not Which)
Answer: We thank the reviewer for yourkindness. It wasrevised accordingly. Thank you.
Question 4: -line 88: increased is not correct. Revise the phrase.
Answer: We thank the reviewer for yourcareful checking. We have revised the description in the discussion (lines 88).
Question 5: -line 141-143: this phrase is not clear to me.
Answer: We have revised the description in the discussion (lines 141-143)“The upregulation of Galand endocannabinoid systems supported the hypothesis of their neuroprotective roles, which are established prior to the onset of clear clinical cognitive symptoms of the disease.”Thank you
Question 5: I would recommend putting the general conclusions at the very end of the article and not before the two specific commentaries on DEGs.
Answer: We thank the reviewer for yourkind suggestion. We have revised the description in the discussion accordingly. Thank you.
Thank you again for your kind and thoughtful comments. We hope that the revision addresses you concerns.
Yours sincerely,
Guo-Li Song
On behalf of all authors
College of Life Sciences and Oceanography
Shenzhen University
Shenzhen, 518071
P.R. China